# EvoSchema: Towards Text-to-SQL Robustness against Schema Evolution

## Abstract

Neural text-to-SQL models, which translate natural language questions (NLQs) into SQL queries given a database schema, have achieved remarkable performance. However, database schemas frequently evolve to meet new requirements. Such schema evolution often leads to performance degradation for models trained on static schemas. Existing work either mainly focuses on simply paraphrasing some syntactic or semantic mappings among NLQ, DB and SQL or lacks a comprehensive and controllable way to investigate the model robustness issue under the schema evolution. In this work, we approach this crucial problem by introducing a novel framework, `EvoSchema`, to systematically simulate diverse schema changes that occur in real-world scenarios. `EvoSchema` builds on our newly defined schema evolution taxonomy, which encompasses a comprehensive set of eight perturbation types, covering both column-level and table-level modifications. We utilize this framework to build an evaluation benchmark to assess the models' robustness against different schema evolution types. Meanwhile, we propose a new training paradigm, which augments existing training data with diverse schema designs and forces the model to distinguish the schema difference for the same questions to avoid learning spurious patterns. Our experiments demonstrate that the existing models are more easily affected by table-level perturbations than column-level perturbations. In addition, the models trained under our paradigm exhibit significantly improved robustness, achieving up to 33 points improvement on the evaluation benchmark compared to models trained on unperturbed data. This work represents a significant step towards building more resilient text-to-SQL systems capable of handling the dynamic nature of database schemas.[1]

## 1 Introduction

Text-to-SQL parsing aims to translate natural language questions (NLQs) into SQL queries given a database schema, enabling the development of natural language interfaces that allow users to query data and invoke services without requiring programming skills (Wang et al., 2020; Zhang et al., 2024a; Yu et al., 2018; Zhang et al., 2023; Li et al., 2024; Tai et al., 2023). Existing neural text-to-SQL models have achieved remarkable performance on existing benchmarks (Li et al., 2024; Yu et al., 2018), which play an important role in empowering different platforms such as business and marketing platforms (Song et al., 2024; Zhang et al., 2024b) and being integrated into virtual assistants to enable real-time data query and analysis (Deksne & Skadiņš, 2022).

However, database schemas are not static; they frequently evolve to accommodate new use cases and improve efficiency (Hillenbrand & Störl, 2021; Cleve et al., 2015). For instance, depending on the scenario, a large patient table might be merged from or split into two tables: a patient information table and a patient diagnosis table (Figure 1-c), to reduce redundancy, enhance data integrity, and optimize performance (Kumar & Azad, 2017). Such schema evolution occurs frequently, which often leads to distribution shifts (Quionero-Candela et al., 2009; Koh et al., 2021) such as nomenclature shifts, data granularity shifts, table and column relation shifts and schema complexity shifts. These distribution shifts can cause significant performance degradation when the model trained on old database schema is adapting to new schema designs.

---

[1]Our code and data will be publicly available.

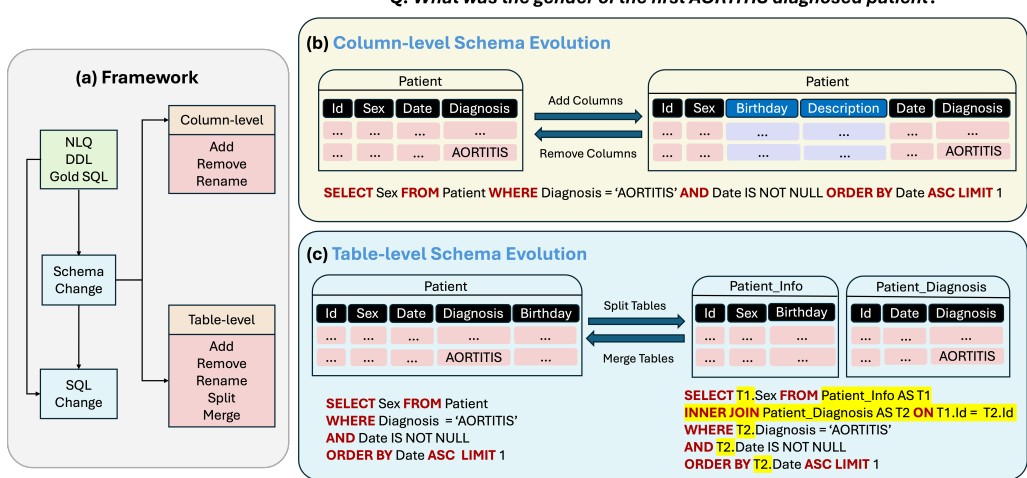

Figure 1: The left (a) is the overview of the `EvoSchema` framework. The top right (b) is a column-level schema evolution example; the bottom right (c) is a table-level schema evolution example.

This challenge highlights a crucial issue in model robustness: how well can a text-to-SQL model adapt to changes in the database schema? Recent studies introduce evaluation benchmarks designed to expose robustness issues by perturbing NLQs, databases or SQL queries (Chang et al., 2023; Deng et al., 2021; Pi et al., 2022; Ma & Wang, 2021). However, these studies have at least one of the following limitations: 1) mainly focus on the syntactic paraphrasing or simple semantic mappings among NLQ, DB and SQL (Chang et al., 2023; Deng et al., 2021); (2) lack a taxonomy of comprehensive schema evolution types (Pi et al., 2022); (3) only focus on schema evolution that does not lead to SQL changes (Ma & Wang, 2021). These efforts are insufficient in the face of increasingly complex and rich database schema changes found in reality. Meanwhile, while it is natural to consider collecting new data after schema evolution for retraining a model, repeating the entire model training life cycle frequently can be costly in terms of both time and resources.

Under this background, we seek to answer the following two questions: (1) How sensitive are existing text-to-SQL models to various types of database schema changes? (2) How can we train a more robust text-to-SQL model that not only performs well on existing database schemas but also adapts effectively to schema changes? Towards this end, we propose a novel schema evolution synthesis framework, `EvoSchema`, which can simulate a wide range of realistic schema design changes by perturbations. Our framework can augment the existing datasets with more comprehensive and realistic schema change types in a systematic way, which not only builds the foundation to evaluate the robustness against different granularities of schema evolution, but also improves models' ability by forcing models to distinguish the structure difference within the schema so as to avoid learning the spurious patterns.

As illustrated in Figure 1, `EvoSchema` framework builds upon our newly defined taxonomy, which encompasses a total of eight types of perturbations over schema, covering both column-level and table-level changes. Column-level perturbations include adding, removing, and renaming columns, while table-level perturbations involve adding, removing, renaming, splitting, and merging tables. We keep the NLQs fixed and examine the robustness of a model under different schema evolutions, and show that existing models are more easily affected by table-level perturbations than column-level perturbations. Moreover, we enhance model robustness by training them with the same questions but coupled with different schema designs to generate the corresponding SQL queries. This training procedure forces the model to distinguish the schema difference which can help models gain a stronger ability to recognize the correct table and column relation and map them to the questions. Our experimental results demonstrate that the perturbation data generated by this framework can help train better text-to-SQL models, which are more robust to different schema evolution types, especially on table-level perturbations.

In summary, our main contributions are as follows:

- We formulate a crucial schema evolution adaptive text-to-SQL problem and present a novel framework, `EvoSchema` to study this problem. We introduce a comprehensive taxonomy of the schema evolution types and build the framework based on the taxonomy to synthesize realistic schema designs by column-level and table-level perturbations.

- We develop an evaluation benchmark that allows for thorough and comprehensive assessment of model robustness against various schema perturbations.

- We propose a new training paradigm: augmenting the existing training data with different schema designs, which not only increase the data diversity, but also force the model to distinguish the schema difference during training. Our approach yields better text-to-SQL models that achieve up to 33 points gain on different types of schema perturbation evaluation data, compared to models trained on unperturbed, original training data.

## 2 METHODOLOGY

### 2.1 BACKGROUND

In the dynamic landscape of databases, schemas frequently evolve to meet new demands, introducing significant challenges for text-to-SQL models (Delplanque et al., 2018; Cleve et al., 2015). These schema changes can vary widely, from minor modifications to complete restructuring, and can significantly impact the performance of models trained on static schemas. In realistic scenarios, a database can often contain a large number of tables, and only several related tables are responsible for a natural language question (NLQ). In our experiment, we represent the relevant database schema using Data Definition Language (DDL) [2] and combine it with the NLQ as input. This input is then used to prompt the model to generate the corresponding SQL query.

### 2.2 RATIONALE FOR SCHEMA EVOLUTION TYPES

When a database schema evolves, it can induce distribution shifts in the data that may impact model performance. We categorize potential distribution shifts into four types: nomenclature shifts, data granularity shifts, table and column relation shifts, and schema complexity shifts. (1) Nomenclature shifts occur when tables and columns are renamed, which may alter the convention of the established terminology within the schema. For example, tables originally named "Products", "Customers", and "Orders" might be renamed to "Items", "Clients", and "Purchases", respectively. Such changes often reflect updates in business terminology or compliance with new standards. A desired model should handle those nomenclature shifts to adapt to the new terminology. (2) Data granularity shifts arise from adding or removing columns or tables, which changes the level of detailedness captured in the database. For instance, an "Employee" table with a single "ContactNumber" field might involve another two separate "WorkContact" and "PersonalContact" fields later. This increases the data granularity to meet new requirements, necessitating models to adapt to more complex and detailed semantics. (3) Table and column relation shifts and schema complexity shifts mainly result from restructuring tables through splitting or merging. This process can highly affect how each table is related to other tables by which column. Both the primary keys and foreign keys may change along with the table restructure. Besides, the schema complexity may change when multiple tables merge from or split into one table. A desired model is expected to be robust to such changes. By categorizing the distribution shifts caused by schema evolution, we can more effectively understand and evaluate a model's capacity to adapt to changes in the underlying database schema.

### 2.3 SCHEMA EVOLUTION SYNTHESIS FRAMEWORK

Our study aims to cover comprehensive potential schema evolution types, which can foster the robustness evaluation of the existing text-to-SQL models and inspire robust model training. We synthesize all the schema evolution types through hybrid strategies, which will leverage both the heuristic rules to guarantee the data quality and LLMs to ensure diversity.

---

[2]DDL defines the structure and properties of a database, providing detailed information necessary for database creation, including column types and primary/foreign keys.

**Broad Coverage of Different Schema Evolution Types:** We aim to encapsulate a broad range of schema evolution types, recognizing their prevalence and impact in real-world scenarios. Specifically, our schema evolution taxonomy includes both column-level and table-level perturbations, which are categorized into eight distinct types. Column-level perturbations comprise three types: adding, removing, and renaming columns, where modifications are restricted to the columns within existing tables. Table-level perturbations encompass five types: adding, removing, renaming, splitting, and merging tables. These perturbations occur frequently in practice, underscoring the need for text-to-SQL models that can robustly handle such changes.

**Hybrid Data Synthesis Strategies:** To ensure both diversity and quality in the generation of schema perturbations, we employ a combination of heuristics and GPT models to synthesize various perturbation types. For each given seed instance, consisting of a *<NLQ, relevant schema, SQL>* triple, we maintain the natural language question (NLQ) fixed across all perturbation types, while only modifying the relevant schema. The corresponding SQL query is adjusted as necessary to remain consistent with the changes in the database schema.

## 2.4 DATA GENERATION

Our proposed schema evolution framework can simulate different types of schema perturbations in a configurable way. For adding or renaming columns, both the modified column size and the column position in the tables are set randomly, and we set the original column size in the table as the maximum number of columns to be changed. For removing columns, we can randomly remove important or unimportant columns from the existing relevant tables. The important columns are the columns that appear in the gold SQL, which will inevitably affect the prediction. For adding, removing, or renaming tables, we randomly add, remove or rename one or multiple tables.

**Schema Change:** To ensure the diversity and reasonability of the synthesized schema, we leverage the capabilities of GPT-3.5 and GPT-4 to synthesize realistic and contextually appropriate columns or tables, which help effectively produce high-quality synthetic data that meets our requirements. For adding or renaming columns and tables, we input the existing relevant tables to GPT-3.5, and let the model generate the potential tables or columns that fit the context. For splitting tables or merging tables, since they are more complex than other perturbations, we use GPT-4 to choose the tables that can be split or merged and then use the modified tables to replace the original ones. For adding or renaming columns and tables, we apply heuristics to choose the suitable synthesized tables or columns, which are not duplicated with the existing ones. Besides, to ensure the correct relationship among different tables after modifying the schema, we apply heuristics to ensure all the foreign keys change along with their referenced table names and column names. When removing columns or tables, any foreign keys in other tables that reference the removed columns or tables will be removed as well.

**SQL Change:** To ensure the consistency of the *<NLQ, relevant schema, SQL>*, after we change the relevant table schema, we revise the gold SQL accordingly. Since the NLQs are the same for adding or removing columns and tables, and the schema evolution here doesn't affect answering the questions, we keep the gold SQL unchanged for these perturbation types. For renaming columns or tables, we revise gold SQL if they appear in the gold SQL. For table splitting or merging, due to the complexity and variation in the required SQL changes, we use GPT-4 to revise the gold SQL. This revision is based on the mappings from the original to the new tables and columns, as well as the necessary adjustments to the JOIN paths. We manually check the edited gold SQL for the evaluation benchmark to make sure they are correct.

By employing these strategies, `EvoSchema` offers a comprehensive and diverse set of schema evolution scenarios that mirror the complexities encountered in real-world database management. By integrating heuristics with LLM-generated perturbations, we maintain both of the diversity and quality, ensuring that the synthesized data is both realistic and challenging.

## 2.5 TRAINING PARADIGM

In our work, we propose a new training paradigm to enhance the model's robustness against different schema evolution. For each *<NLQ, relevant schema, SQL>* triple, we fix the NLQ in the training data, and augment each triple with different schema designs, which may or may not lead to SQL

change. Consequently, we obtain multiple triples that can be derived from each of the original triples. We train the model by learning multiple schema designs and SQLs to the original question mappings, which can improve the model's ability to identify the correct relationships among different tables and columns to the question, and can better distinguish the difference among different schema designs. Through this procedure, the model can avoid learning spurious patterns better and therefore enhance the robustness against different schema evolution types.

## 3 EXPERIMENT SETUP

### 3.1 DATASET

For our experiments, we utilize the BIRD (Li et al., 2024) and Spider (Yu et al., 2018) datasets, which are specifically designed for the text-to-SQL task. Both of them consist of NLQs, corresponding database schemas, and gold SQL queries. These datasets are diverse, encompassing a wide range of real-world database scenarios, which provides a robust foundation for evaluating the performance of models in translating NLQs into SQLs.

Schema Perturbations: To evaluate the robustness of the text-to-SQL models, we use the BIRD and Spider datasets not only in their original form but also augmented with various column-level and table-level schema perturbations. We ensure that the NLQs remain fixed, while the schema and SQL queries are adjusted as necessary to reflect the changes introduced by our perturbations. We follow the standard train/dev split provided with these datasets, and apply all the perturbations on both training data and evaluation data. The data statistics are in Table 8 and the examples of different perturbation types are in Figure 2 in the Appendix.

### 3.2 TRAINING AND EVALUATION SETTINGS

**Training Setting:** We choose four open-source models: Code Llama-7B (Rozière et al., 2024), Mistral-7B (Jiang et al., 2023), Llama 3-8B (Dubey et al., 2024) and SQLCoder-7B [3] and two closed-source models: GPT-3.5 [4] and GPT-4 (OpenAI et al., 2024) for our experiments. For these four open-source models, we explore two settings: 1) without perturbation types: the model is trained on the original training data without any perturbation types introduced during training. 2) with perturbation types: the model is trained by merging both the original training data and the perturbation training data. For closed-source models, we only use them for evaluation.

**Evaluation Setting:** For all the closed-source models and the finetuned open-sourced models, we evaluate them under two settings: 1) without perturbation types: this setting uses the standard, unaltered original evaluation data to evaluate the model performance. 2) with perturbation types: the models are evaluated on data where different perturbations are introduced. By comparing the model performance under these two settings, we can assess how resilient the finetuned models and GPT models are to schema evolution in text-to-SQL parsing. This setup provides a comprehensive evaluation of model performance in both standard and perturbed environments, allowing for detailed analysis of robustness and adaptability across different models and schema evolution types.

### 3.3 EVALUATION METRICS

1) Table Match F1: this score is a metric to measure how well the model correctly identifies the relevant tables required to generate a valid SQL query. The F1 score is a harmonic mean of precision and recall, where the precision is the percentage of tables correctly predicted out of all tables predicted by the model and the recall is the percentage of tables correctly predicted out of all the actual tables that should have been selected. The Table Match F1 score combines these two metrics to provide a balanced evaluation, which can assess the ability of text-to-SQL models to correctly identify the required tables from the database schema to form accurate queries. A higher Table Match F1 indicates better performance in selecting the correct tables for the SQL query.

2) Column Match F1: this score is to evaluate how accurately the model identifies the relevant columns required to generate a valid SQL query from a natural language input. Like the Table

---

[3]https://huggingface.co/defog/sqlcoder-7b-2

[4]https://openai.com/chatgpt/

Match F1, it measures the balance between precision and recall but is applied specifically to the columns of the database. A higher Column Match F1 score indicates better performance in selecting the right columns for the SQL query.

3) Execution Accuracy: this metric measures whether the predicted SQL query can return the correct results as the gold SQL when executing against a database. Since the schema evolution may lead to database restructure and there are no existing values for the new database after schema change, we synthesize values to create new databases and execute the new gold SQLs after schema evolution on them. Due to the complexity of the value synthesis and huge manual efforts to ensure an executable database for each instance, we filter out the cases where synthesized database is not executable by new gold SQL. This procedure can lead to very small size of the evaluation data for some perturbation types, so we mainly use the other two metrics as the main metrics.

### 3.4 TRAINING AND EVALUATION DETAILS

We choose Code Llama-7B (Rozière et al., 2024), Mistral-7B (Jiang et al., 2023), Llama 3-8B (Dubey et al., 2024) and SQLCoder-7B [3] as our open-source base models. We fine-tune these models with Huggingface transformers library (Wolf et al., 2020). For the perturbation training, We merge all the perturbation data and randomly shuffle them as our final training data. We use a learning rate of 2e-5 for training Code Llama, Llama 3 and SQLCoder, and 5e-6 for training Mistral. Our batch size is 4. We train all the models on 4 A100 80GB GPUs and use a cosine scheduler with a 0.03 warm-up period for 6 epochs. We employ FSDP (Zhao et al., 2023) to efficiently train the model. We set the max input length of training as 1024 and the max output length of inference as 500. For inference, we use vllm (Wolf et al., 2020) for batch evaluation, and we set the batch size as 16. We do the inference on an 80G A100 GPU. For closed-source LLMs, we use Azure OpenAI API[5]. We use the 2023-12-01-preview version for GPT-4, and 2023-07-01-preview version for GPT-3.5.

## 4 RESULTS AND ANALYSIS

### 4.1 MAIN RESULTS

As Table 1 and Table 2 shows, we train Codellama, Mistral, Llama3 and SQLCoder on the original BIRD training data with and without different perturbation types, and evaluate the model on the original BIRD evaluation data and different perturbation types. We observe that:

**The models trained on different perturbation types are more robust to the schema variation.** Adding the perturbation data during training: 1) does not sacrifice the performance of the original evaluation data; 2) achieves comparable or better results on different perturbation types. Column-level schema changes are relatively minor compared with table-level schema changes. We can see the models perform better on both the column-level and table-level perturbation types in general, which shows the models are robust to both minor schema evolution and major schema evolution.

**The models trained on different perturbation types demonstrate high robustness on the table-level schema evolution.** Adding the perturbation data during training achieves significantly better results on table-level perturbation types (i.e., major schema change types). By comparing these four models' performance with and without the perturbation data, we observe that for adding tables, the model trained with perturbation data can achieve up to 33 points improvement for table match F1 and 18 points improvement for column match F1; for splitting tables, the model trained with perturbation data can achieve up to 14 points improvement for table match F1 and 4.2 points improvement for column match F1; for merging tables, the model trained on perturbation data can achieve up to 4 points improvement on table match F1 and 3 points improvement for column match F1.

**Closed-source models are robust to different scheme evolution types in general.** As table 1 shows, we compare the model performance on GPT models and four open-source models trained with and without perturbation types. We observe that: the GPT models are robust to different schema evolution types in general, which can have much better results than the models trained without perturbation types. Besides, even for those major schema change types such as adding, splitting and merging tables, the GPT results are still very close to the performance compared with

---

[5]https://learn.microsoft.com/en-us/azure/ai-services/openai/reference

Table 1: Evaluation on BIRD. "w/": the model is trained by merging the original data and all the perturbation training types together; "w/o": the model is only trained on the original training data. The **best performance** for each model is in bold, and red shows a larger gain. "-": some of the relevant tables are removed so there should be no gold SQL used to calculate the metrics here.

| Perturbation Type | Code Llama | | Mistral | | Llama 3 | | SQLCoder | | GPT-3.5 | GPT-4 |
| --- | --- | --- | --- | --- | --- | --- | --- | --- | --- | --- |
| | w/o | w/ | w/o | w/ | w/o | w/ | w/o | w/ | | |
| Table Match F1 | | | | | | | | | | |
| Original | 89.77 | **90.42** | 89.58 | **90.62** | **89.96** | 89.53 | 89.69 | **90.64** | 87.28 | 88.98 |
| Add Columns | 89.73 | **90.27** | 89.65 | **90.03** | 89.08 | **89.70** | 89.30 | **90.52** | 86.35 | 88.12 |
| Remove Columns | 89.82 | **90.24** | 89.89 | **90.66** | **90.09** | 89.82 | 89.81 | **90.54** | 87.18 | 88.87 |
| Rename Columns | **85.28** | 85.07 | **84.32** | 84.27 | **83.74** | 82.92 | **85.32** | 84.93 | 81.73 | 83.20 |
| Add Tables | 57.88 | **89.50** | 57.67 | **89.30** | 55.11 | **88.51** | 57.44 | **89.38** | 83.54 | 85.79 |
| Remove Tables | - | - | - | - | - | - | - | - | - | - |
| Rename Tables | 88.84 | **90.32** | 89.40 | **90.56** | 87.18 | **89.14** | 89.40 | **90.48** | 87.02 | 88.45 |
| Split Tables | 71.99 | **81.55** | 66.12 | **80.87** | 71.08 | **80.12** | 72.52 | **81.92** | 77.52 | 80.68 |
| Merge Tables | 87.52 | **88.95** | 85.52 | **88.50** | 83.88 | **87.82** | 86.70 | **88.13** | 84.88 | **87.09** |
| Column Match F1 | | | | | | | | | | |
| Original | 80.66 | **81.64** | 81.10 | **82.36** | **79.13** | 78.72 | 81.52 | **81.97** | 78.28 | 80.78 |
| Add Columns | 78.26 | **80.27** | 79.16 | **80.18** | 75.79 | **76.87** | 79.09 | **80.46** | 75.03 | 78.58 |
| Remove Columns | 82.67 | **82.75** | 83.09 | **84.00** | **81.56** | 80.69 | **83.20** | 83.18 | 80.33 | 82.55 |
| Rename Columns | 76.50 | **76.94** | 76.35 | **76.73** | **72.24** | 71.07 | 76.84 | **77.38** | 73.40 | 75.90 |
| Add Tables | 63.81 | **81.14** | 65.39 | **81.09** | 59.36 | **77.96** | 62.91 | **81.23** | 76.45 | 79.32 |
| Remove Tables | - | - | - | - | - | - | - | - | - | - |
| Rename Tables | 79.60 | **80.91** | 80.32 | **81.29** | **77.49** | 77.46 | 80.77 | **81.79** | 77.78 | 80.04 |
| Split Tables | 75.30 | **78.45** | 73.87 | **78.11** | 73.81 | **73.95** | 75.83 | **78.59** | 74.89 | 77.41 |
| Merge Tables | 67.73 | **68.98** | 66.27 | **69.39** | 65.60 | **65.79** | 67.55 | **69.09** | 65.07 | 69.13 |

the original evaluation data. *However, comparing the model performance on the open-source LLMs and closed-source LLMs, the models trained with perturbation data have better performance than GPT models on both column-level perturbation and column-level perturbation evaluation data.* This indicates that our models trained with perturbation data are more robust than GPT models.

**Table-level perturbation has a larger impact than column-level perturbation on the model performance.** As Table 1 shows, comparing with the performance on the original evaluation data: adding tables and splitting tables will lead to a significant table match F1 drop; adding tables, splitting tables and merging tables will lead to a significant column match F1 drop. This phenomenon indicates that adding tables or splitting tables easily confuses the models in choosing the correct tables to generate the SQL query. For merging tables, even though the model can correctly choose tables, it's a bit hard for the model to pick up the correct columns when the columns from different tables go into the same table. While for the column-level performance, there are limited differences with the performance on the original data.

**Reducing table schema complexity is beneficial for model performance.** Compare the model performance on column-level perturbation evaluation and the original evaluation data, adding columns results in a decrease in column match F1, whereas removing columns leads to an increase in column match F1. It indicates simpler table schema is beneficial for models to select columns, as removing columns simplifies the table schema while adding columns makes the table schema more complex.

## 4.2 INFLUENCE OF IRRELEVANT TABLES

We observed that the model trained with perturbation types demonstrates significant robustness to table-level perturbations, such as adding and splitting tables. Upon analyzing the errors, we found that models trained without perturbation types tend to predict SQL queries that join all available tables, even when some tables are irrelevant to the NLQs and SQLs. We hypothesize that this occurs because during training without perturbations, the model only sees relevant table schemas, causing it to learn spurious patterns that always try to join all the input tables.

Table 2: Evaluation on BIRD. "w/": the model is trained with all the perturbation types; "w/o": the model is only trained on the original training data.

Table 3: Irrelevant tables effect. "w/": the model is trained with all the perturbation types; "w/o": the model is only trained on the original training data; "w/o+": the model is only trained on the original training data, but for the input table schema, we also add irrelevant tables.

| Exec Acc on BIRD | | |
|---|---|---|
| Perturbation Type | Exec Acc | |
| | w/o | w/ |
| Original | **62.36** | 61.47 |
| Add Columns | **60.99** | 60.31 |
| Remove Columns | **62.61** | 60.67 |
| Rename Columns | 58.75 | **58.90** |
| Add Tables | 49.50 | **62.54** |
| Remove Tables | - | - |
| Rename Tables | 58.64 | **59.05** |
| Split Tables | 50.00 | **60.67** |
| Merge Tables | 48.76 | **52.88** |

| Add Irrelevant Tables Effect | | | | | | |
|---|---|---|---|---|---|---|
| Perturbation Type | Table Match F1 | | | Column Match F1 | | |
| | w/o | w/o+ | w/ | w/o | w/o+ | w/ |
| Original | 89.77 | 87.65 | **90.42** | 80.66 | 79.24 | **81.64** |
| Add Columns | 89.73 | 86.35 | **90.27** | 78.26 | 75.31 | **80.27** |
| Remove Columns | 89.82 | 87.30 | **90.24** | 82.67 | 80.74 | **82.75** |
| Rename Columns | **85.28** | 81.90 | 85.07 | 76.50 | 73.28 | **76.94** |
| Add Tables | 57.88 | 88.01 | **89.50** | 63.81 | 79.51 | **81.14** |
| Remove Tables | - | - | - | - | - | - |
| Rename Tables | 88.84 | 86.84 | **90.32** | 79.60 | 78.47 | **80.91** |
| Split Tables | 71.99 | 67.27 | **81.55** | 75.30 | 70.39 | **78.45** |
| Merge Tables | 87.52 | 85.36 | **88.95** | 67.73 | 65.78 | **68.98** |

To explore whether simply adding irrelevant tables could yield similar performance to models trained with perturbation data, we conducted an experiment where we trained Code Llama on BIRD. As shown in Table 3, adding irrelevant tables led to similar performance on the "Add Tables" perturbation type. However, it caused a performance drop on other perturbation types. This suggests that combining all perturbation data is necessary to train a more robust model.

## 4.3 INFLUENCE OF PERTURBATION TYPES

We explore the effect of the column-level perturbation types and table-level perturbation types. As Table 4 shows, we train the model with both column-level and table-level perturbation types, and compare it with the model trained without column-level perturbation types and without table-level perturbation types. From our experiments, we found that without training on table-level perturbations, the model performance can be slightly better than the model trained with both column-level and table-level perturbation types on column-level perturbation types, while can lead to a significant performance drop on the table-level perturbation types. This indicates that the table-level perturbation data has a limited effect on the column-level perturbation types while having a huge impact on the table-level perturbation types. When looking at the model trained only on table-level perturbation types, we found that the model performance on both column-level and table-level perturbation types dropped. This indicates that the column-level perturbation types can still benefit the training.

## 4.4 INFLUENCE OF OUT-OF-SCOPE TYPES

In our experiments, we investigate both in-scope and out-of-scope scenarios. For in-scope experiments, the gold SQL query may or may not change in response to modifications in the database schema. In contrast, out-of-scope experiments involve two special perturbation types: 1) *Removing columns that appeared in gold SQL:* columns that are present in the gold SQL are removed from the schema. 2) *Removing tables that appeared in gold SQL:* tables that are referenced in the gold SQL are removed from the schema. Here, we anticipate that the model refuses to generate SQL because the provided column information (the former) and table information (the latter) are insufficient.

To evaluate the impact of these perturbations, we incorporate the two out-of-scope perturbation types into the training data, along with the original training data and all other in-scope perturbation types. We then compare the performance of a model trained on this combined dataset against models trained solely on the original training data and models trained only with in-scope perturbation data. From Table 5, we can see that with out-of-scope perturbation training data, the model performance drops on the original evaluation data and all the other in-scope perturbation evaluation data. By analyzing the errors, we found that the model tends to make more conservative predictions, which will refuse to predict SQL sometimes for the cases where the gold SQL exists. We further analyze the false positive (FP) and true positive (TP) under the model trained with out-of-scope perturbation

Table 4: Perturbation type ablation on BIRD. The base model is Code Llama. "both": the model is trained with both column-level perturbation and table-level perturbation types; "w/o table-p": the model is trained without table-level perturbation types; "w/o column-p": the model is trained without column-level perturbation types.

| Perturbation Type | Table Match F1 | | | Column Match F1 | | |
|---|---|---|---|---|---|---|
| | both | w/o table-p | w/o column-p | both | w/o table-p | w/o column-p |
| Original | 90.73 | 90.80 (+0.07) | 90.04 (-0.69) | 81.09 | 82.15 (+1.06) | 80.49 (-0.60) |
| Add Columns | 90.86 | 90.80 (-0.06) | 89.75 (-1.11) | 79.63 | 80.81 (+1.18) | 77.29 (-2.34) |
| Remove Columns | 90.72 | 90.83 (+0.11) | 90.48 (-0.24) | 83.28 | 83.85 (+0.57) | 82.61 (-0.67) |
| Rename Columns | 85.35 | 85.38 (+0.03) | 84.57 (-0.78) | 76.49 | 77.53 (+1.04) | 75.17 (-1.32) |
| Add Tables | 88.95 | 58.94 (-30.01) | 88.57 (-0.38) | 79.87 | 64.11 (-15.76) | 79.33 (-0.54) |
| Remove Tables | - | - | - | - | - | - |
| Rename Tables | 90.54 | 90.77 (+0.23) | 89.29 (-1.25) | 81.13 | 81.51 (+0.38) | 79.33 (-1.80) |
| Split Tables | 80.71 | 73.28 (-7.43) | 79.05 (-1.66) | 77.41 | 75.95 (-1.46) | 76.30 (-1.11) |
| Merge Tables | 88.72 | 87.87 (-0.85) | 86.83 (-1.89) | 68.40 | 68.26 (-0.14) | 67.08 (-1.32) |

Table 5: Out of Scope Effect on BIRD. The base model is Code Llama. "w/o": the model is trained without perturbation types; "w/": the model is trained on the original data and all the perturbation types; "+ OOS": the model is trained on the original data, perturbation types and two out-of-scope (OOS) perturbation types; "+ OOS FP": The model trained with two out-of-scope perturbation types makes an incorrect prediction on the original data and in-scope perturbation data; "+ OOS TP": The model trained with two out-of-scope perturbation types makes the correct prediction on the two out-of-scope perturbation data; "Tab": the model refuses to predict SQL due to the lack of table information; "Col": the model refuses to predict SQL due to the lack of column information.

| Perturbation Type | Table Match F1 | | | Column Match F1 | | | + OOS FP | | + OOS TP | |
|---|---|---|---|---|---|---|---|---|---|---|
| | w/o | w/ | + OOS | w/o | w/ | + OOS | Tab | Col | Tab | Col |
| Original | 89.77 | 90.42 | 82.98 (-7.44) | 80.66 | 81.64 | 75.43 (-6.21) | 7.11 | 0.65 | - | - |
| Add Columns | 89.73 | 90.27 | 86.07 (-4.20) | 78.26 | 80.27 | 77.00 (-3.27) | 4.25 | 0.40 | - | - |
| Remove Columns | 89.82 | 90.24 | 82.24 (-8.00) | 82.67 | 82.75 | 75.90 (-6.85) | 7.56 | 0.72 | - | - |
| Remove Col in SQL | - | - | - | - | - | - | 5.02 | - | - | 84.03 |
| Rename Columns | 85.28 | 85.07 | 80.20 (-4.87) | 76.50 | 76.94 | 73.04 (-3.90) | 4.44 | 0.20 | - | - |
| Add Tables | 57.88 | 89.50 | 88.78 (-0.72) | 63.81 | 81.14 | 80.71 (-0.37) | 0.33 | 0.07 | - | - |
| Remove Tables | - | - | - | - | - | - - | - | 1.62 | 83.86 | - |
| Rename Tables | 88.84 | 90.32 | 86.36 (-3.96) | 79.60 | 80.91 | 78.06 (-2.85) | 3.52 | 0.39 | - | - |
| Split Tables | 71.99 | 81.55 | 81.07 (-0.48) | 75.30 | 78.45 | 78.02 (-0.43) | 0.26 | 0.07 | - | - |
| Merge Tables | 87.52 | 88.95 | 83.85 (-5.10) | 67.73 | 68.98 | 65.42 (-3.56) | 4.65 | 0.35 | - | - |

types, the FP is very close to the gap between the model trained with and without out-of-scope perturbation types, which can help verify that the model becomes more conservative to the response is the major reason to lead the performance drop. Besides, we found that by removing columns in gold SQL and removing tables, the TP is only around 84%, which indicates that the model still has a 16% chance to make a prediction even when there should not be an SQL.

## 4.5 Influence of Intra-database and Cross-database

We hypothesize that a model trained on the same databases may not only learn schema evolution patterns but also become familiar with specific table and column names. To test this, we split the BIRD training data into train/test sets to ensure that each database in the test set also appears in the training set. We use Code Llama as the base model. The results in Table 6 show that, for most perturbation types, the model's performance improves more compared to the cross-database scenario in Section 4.1, which verifies our hypothesis.

Table 6: Intra-database Effect. This experiment emphasizes that the training and evaluation occur within the same database, instead of across databases.

| Intra-database Effect | | | | |
|---|---|---|---|---|
| Perturbation Type | Table Match F1 | | Column Match F1 | |
| | w/o | w/ | w/o | w/ |
| Original | 87.24 | **87.43** | 79.54 | **80.89** |
| Add Columns | 87.14 | **87.43** | 76.36 | **78.92** |
| Remove Columns | **87.29** | 87.27 | 81.14 | **81.29** |
| Rename Columns | 85.71 | **86.43** | 77.45 | **79.09** |
| Add Tables | 61.13 | **83.95** | 66.11 | **78.57** |
| Remove Tables | - | - | - | - |
| Rename Tables | 86.33 | **86.67** | 79.44 | **79.96** |
| Split Tables | 71.82 | **78.52** | 75.09 | **77.42** |
| Merge Tables | 85.11 | **87.44** | 71.43 | **74.72** |

Table 7: Evaluation on Spider. "w/": the model is trained by merging the original data and all perturbation types; "w/o": the model is only trained on the original training data.

| Spider Evaluation | | | | |
|---|---|---|---|---|
| Perturbation Type | Table Match F1 | | Column Match F1 | |
| | w/o | w/ | w/o | w/ |
| Original | **99.72** | 99.65 | 90.54 | **91.35** |
| Add Columns | **99.73** | 99.66 | 88.81 | **90.94** |
| Remove Columns | 99.35 | **99.65** | 86.13 | **88.59** |
| Rename Columns | 99.68 | **99.74** | 86.55 | **89.13** |
| Add Tables | 62.46 | **98.80** | 66.49 | **90.54** |
| Remove Tables | - | - | - | - |
| Rename Tables | 99.24 | **99.25** | 89.71 | **90.99** |
| Split Tables | 73.27 | **90.24** | 76.76 | **86.93** |
| Merge Tables | 96.03 | **98.34** | 78.03 | **83.43** |

## 4.6 GENERALIZABILITY TO OTHER DATASETS

To evaluate the generalizability of `EvoSchema` to other text-to-SQL datasets, we conducted experiments on the Spider dataset and used Mistral as the base model. As shown in Table 7, we reached conclusions consistent with those in Section 4.1, which further demonstrates the effectiveness and utility of our proposed framework and training methods.

## 5 RELATED WORK

Existing research on text-to-SQL robustness is mainly two-fold: robustness evaluation and robustness training. Recent studies introduce evaluation benchmarks designed to expose robustness issues by perturbing NLQs, databases or SQL queries. However, these studies tend to focus on syntactic paraphrasing or simple semantic mappings, such as different representations of numbers or name abbreviations across NLQ, DB, and SQL (Chang et al., 2023; Deng et al., 2021). While some work analyzes schema changes, they mainly focus on irrelevant column modifications that do not affect SQL (Ma & Wang, 2021) or with limited perturbation types (Pi et al., 2022). These efforts are insufficient in the face of increasingly complex and rich database schemas found in modern datasets. Moreover, the advent of LLMs has mitigated many linguistic challenges, further emphasizing the need for robust adaptation to structural changes in database schemas. For robust training, existing methods employ strategies like decomposing tasks so that models generate each sub-clause individually before merging them (Gao et al., 2022), or using execution-guided decoding to eliminate incorrect sub-clauses (Wang et al., 2018). While these approaches focus on enhancing various aspects of text-to-SQL robustness, our work specifically addresses the challenge of schema evolution.

## 6 CONCLUSION

In conclusion, we formulate the critical challenge of schema evolution in adaptive text-to-SQL systems and introduce `EvoSchema`, a novel framework designed to study and mitigate this problem. We developed a comprehensive taxonomy of schema evolution types, enabling the synthesis of realistic schema designs through column-level and table-level perturbations. Leveraging this taxonomy, we constructed an evaluation benchmark that facilitates thorough and comprehensive assessment of model robustness against various schema perturbations. Furthermore, we proposed a new training paradigm that augments existing training data with diverse schema designs, enhancing data diversity and compelling models to recognize schema differences during training. Our approach significantly improves text-to-SQL models, achieving up to a 33-point gain on various schema perturbation evaluation types compared to models trained on the original, unperturbed data. These findings highlight the effectiveness of our methods in building more robust text-to-SQL models capable of adapting to evolving schemas, paving the way for future advancements in the field.

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

# A APPENDIX

Table 8: Data statistics of original data and perturbation data. "*": the evaluation data for calculating execution accuracy. We synthesize values to reconstruct the database after schema evolution, and filter out those not executable by gold SQL, which results in the smaller size of the evaluation data.

| Data Statistics | | | | | |
|---|---|---|---|---|---|
| Perturbation Type | BIRD | | | Spider | |
| | Train | Eval | Eval* | Train | Eval |
| Original | 9426 | 1534 | 789 | 7000 | 1034 |
| Add Columns | 9219 | 1506 | 582 | 6999 | 1034 |
| Remove Columns | 9426 | 1534 | 773 | 7000 | 1034 |
| Remove Col in SQL | 9424 | 1534 | - | 7000 | 1034 |
| Rename Columns | 9385 | 1533 | 674 | 6979 | 1034 |
| Add Tables | 9387 | 1530 | 606 | 6977 | 1033 |
| Remove Tables | 7212 | 1171 | - | 3069 | 1034 |
| Rename Tables | 9392 | 1534 | 735 | 7000 | 1034 |
| Split Tables | 9254 | 1515 | 178 | 6903 | 1029 |
| Merge Tables | 6930 | 1139 | 402 | 2999 | 437 |

Figure 2: An overview of different perturbation types generated by EvoSchema. The top is an unperturbed example; the middle is the column-level perturbation; the bottom is the table-level perturbation. "Remove Col in SQL": remove columns that appear in gold SQL; "Remove Tables": the relevant tables appear in gold SQL are removed. Thus there is no gold SQL for these two cases.

