# OpenReview forum: "EVOSCHEMA: TOWARDS TEXT-TO-SQL ROBUSTNESS AGAINST SCHEMA EVOLUTION"
_ICLR.cc/2025/Conference — ICLR 2025 Conference Withdrawn Submission_

### Official Review · Reviewer_zbX2 · 2024-10-30

**Soundness:** 2
**Presentation:** 2
**Contribution:** 3
**Rating:** 5
**Confidence:** 4

**Summary:**

Database schemas evolve as end-user information needs change. However, most models are trained on static data and do not account for robustness under schema evolution. This paper addresses this challenge by (1) creating a taxonomy of common schema perturbation types, (2) developing a benchmark that simulates these schema changes, and (3) fine-tuning existing models on training data augmented with these perturbation types.

The paper shows that fine-tuning models on the perturbation-augmented data enhances their robustness to database schema changes.

**Strengths:**

1. This paper addresses a critical problem: enhancing model robustness in the face of data evolution. In the context of Text-to-SQL, data evolution specifically refers to schema changes.
2. The paper develops a taxonomy for schema evolution, mapping each type of perturbation to a real-world scenario.
3. The paper presents a thorough evaluation across a wide range of open-source models as well as state-of-the-art commercial models.

**Weaknesses:**

1. The benchmark generation section is too brief and lacks critical details, which weakens the validity of the paper's results. My main question here is: how do you ensure that adding tables or columns does not create alternative correct SQL answers? Adding a table can introduce alternative join paths or semantically similar columns, which could also be used to answer the NLQ accurately. For example, in Figure 2, when adding an "appointment" table, if this table contains fields such as the patient who makes the appointment, the doctor assigned, and the laboratory records associated with the appointment, it could be used to join "patient" and "laboratory," forming another valid SQL query to answer the question. The paper mentions “applying heuristics to choose suitable synthesized tables or columns, which are not duplicated with existing ones.” However, this is too vague. What are the specific "heuristics"? What defines “suitable”? Is "duplicate" assessed in terms of syntactic similarity or semantic similarity? Please provide a more detailed explanation of the validation process for ensuring the uniqueness of correct answers after schema perturbations.

2. While the evaluation section is detailed, some results are counterintuitive, and the paper does not provide thorough explanations:

    a) In Table 1, why does "adding tables" cause the most significant drop in table match F1? How many tables were added? This seems counterintuitive, as "splitting tables" also increases the number of tables, yet its F1 score is much higher than "adding tables" (without perturbation).

    b) In Table 1, for "adding tables (w/o)," why is the column match F1 higher than the table match F1? This is counterintuitive, as correctly selecting columns typically requires selecting the correct table.

    c) GPT-4 is used to generate the gold query for split and merge table cases. However, in Table 1, its performance is lower than that of models fine-tuned on augmented data for "split/merge tables." This is surprising, as GPT-4 should know the gold query for these scenarios.

Could you provide a more detailed explanation on these counterintuitive results?

**Questions:**

My questions are listed in the second point of weaknesses.

Besides, I have doubts about the explanation in Section 4.2. It states that “during training without perturbations, the model only sees relevant table schemas, causing it to learn spurious patterns that always attempt to join all input tables.” I don’t believe this is accurate, as in BIRD, most gold SQL queries use only 2-3 tables from the database schema and there are many SQLs that only use a single table. It’s unclear to me why the model would learn that joining all tables is a beneficial pattern. An analysis comparing the distribution of table usage in the training data with the table usage distribution in the model-predicted SQLs would be helpful.

---

> ### Author Response · Authors · 2024-11-21
> **Author Response (Part 1/2)**
>
> We sincerely appreciate the reviewer’s recognition of our work in addressing a critical problem, developing a taxonomy that maps schema evolution types to real-world scenarios, and conducting a comprehensive evaluation across a wide range of open-source and closed-source models.
> ***
> [How do you ensure that adding tables or columns does not create alternative correct SQL answers? Adding a table can introduce alternative join paths or semantically similar columns, which could also be used to answer the NLQ accurately.]
>
> For the "adding tables" scenario, the added tables are irrelevant to the question but belong to the same database as the relevant tables in BIRD and Spider. The original BIRD and Spider datasets should ensure that no alternative tables within their databases can produce different but correct SQL answers.
> ***
> [What are the specific "heuristics"? Is "duplicate" assessed in terms of syntactic similarity or semantic similarity? Please provide a more detailed explanation of the validation process for ensuring the uniqueness of correct answers after schema perturbations.]
>
> We take the renaming columns as an example, when renaming columns, the following heuristics should be applied to maintain consistency and avoid conflicts: 1) Preserve Meaning: The new column name should reflect the same meaning as the original column to avoid semantic confusion (e.g., CourseID → ClassId). 2) Avoid Conflicts: Ensure that the new column name does not conflict with existing column names within the same or other tables in the database. 3) Update References: Update all references to the renamed column in foreign keys in other tables. 4) Revise SQL: Update all SQL queries referencing the renamed column to work correctly after the renaming. These heuristics aim to ensure that renaming columns are performed systematically, maintaining the database's integrity and compatibility with SQL queries.
>
> For “duplicated” in adding columns, we first designed the prompts to let GPT models avoid generating column names that have the similar meaning to the column or table names in the existing tables. Then, we also directly use the string match to make sure there are no repeated column names in the table.
> ***
> [In Table 1, why does "adding tables" cause the most significant drop in table match F1? How many tables were added? This seems counterintuitive, as "splitting tables" also increases the number of tables, yet its F1 score is much higher than "adding tables" (without perturbation).]
>
> For adding tables, we randomly add a minimum of 1, an average of 2, and a maximum of 3 tables. For splitting tables, one original table from the gold SQL is split into a minimum of 1, an average of 2.6, and a maximum of 5 tables. As models trained without perturbations only see tables present in the gold SQL during training, the models tend to join most of the given tables during inference. Splitting tables achieves higher performance because most of the new tables still appear in the gold SQL (as shown in the table-level perturbation example in Figure 1). While those added tables in the "adding tables" scenario are irrelevant to the gold SQL, resulting in more errors for adding tables.
> ***
> [In Table 1, for "adding tables (w/o)," why is the column match F1 higher than the table match F1? This is counterintuitive, as correctly selecting columns typically requires selecting the correct table.]
>
> Thank you for your feedback. Could you kindly clarify your question? Specifically, in which scenario or dataset do you observe the column match F1 being higher than the table match F1?

---

> > ### Comment · Reviewer_zbX2 · 2024-11-23
> > **Further Questions**
> >
> > Thanks for providing the clarification.
> >
> > >For the "adding tables" scenario, the added tables are irrelevant to the question but belong to the same database as the relevant tables in BIRD and Spider.
> >
> > Does that mean you do not add new tables outside the original BIRD and Spider benchmark? That confuses me because each question is posed to a certain database in BIRD. What does it mean that you add tables but the added tables belong to the same database?  What is the initial set of tables provided before any are added?
> >
> > >Thank you for your feedback. Could you kindly clarify your question? Specifically, in which scenario or dataset do you observe the column match F1 being higher than the table match F1?
> >
> > In Table 1, your results indicate that the column match F1 score is higher than the table match F1 score.
> >
> > This seems counterintuitive, as column matching is generally more challenging than table matching. Typically, if the columns are correctly identified, the corresponding table should also be correctly identified. The only plausible explanation for the column match F1 being higher than the table match F1 is that columns are unevenly distributed across tables. It appears that the model may accurately identify many columns in certain tables but fails to do so consistently across other tables.

---

> > > ### Author Response · Authors · 2024-11-25
> > > **Further Response**
> > >
> > > Thanks for your questions!
> > >
> > > [Does that mean you do not add new tables outside the original BIRD and Spider benchmark? That confuses me because each question is posed to a certain database in BIRD. What does it mean that you add tables but the added tables belong to the same database? What is the initial set of tables provided before any are added?]
> > >
> > > BIRD has multiple databases. Each database contains multiple tables. Each question is only posed to certain tables within a database in BIRD. So for inference, we can add the tables that are irrelevant to the question but can still be in the same database as those relevant tables. The initial set of the tables provided are those relevant tables to a question within a database.
> > >
> > > [This seems counterintuitive, as column matching is generally more challenging than table matching. Typically, if the columns are correctly identified, the corresponding table should also be correctly identified. The only plausible explanation for the column match F1 being higher than the table match F1 is that columns are unevenly distributed across tables. It appears that the model may accurately identify many columns in certain tables but fails to do so consistently across other tables.]
> > >
> > > Yes. It may appear especially that the columns are unevenly distributed across tables. The F1 score calculation for columns might be more forgiving due to the higher number of columns compared to tables in a query. Missing a single table in a query that requires multiple tables can significantly lower the table match F1 score, whereas correctly predicting most of the columns can keep the column match F1 score high. If a query contains 3 tables and 6 columns, wrongly predicting 1 table can lead to a larger impact than wrongly predicting a column.

---

> ### Author Response · Authors · 2024-11-21
> **Author Response (Part2/2)**
>
> [GPT-4 is used to generate the gold query for split and merge table cases. However, in Table 1, its performance is lower than that of models fine-tuned on augmented data for "split/merge tables." This is surprising, as GPT-4 should know the gold query for these scenarios.]
>
> The GPT-4 results in Table 1 are just slightly lower than the fine-tuned models’ results. Here are two reasons: 1) When synthesizing ground truth new SQLs after schema evolution, we manually correct some wrong SQLs generated by GPT4, which means GPT4 can not always generate correct SQLs; 2) When synthesizing SQLs for “split/merge tables’, we prompt GPT4 to edit the original SQL before “split/merge tables”. This editing task is easier for GPT4 compared with directly generating SQLs. So for the GPT-4 results in Table 1, SQLs are directly generated, leading to a higher likelihood of errors.
> ***
> [Besides, I have doubts about the explanation in Section 4.2. It states that “during training without perturbations, the model only sees relevant table schemas, causing it to learn spurious patterns that always attempt to join all input tables.” It’s unclear to me why the model would learn that joining all tables is a beneficial pattern.]
>
> This is not a beneficial pattern but a phenomenon observed in the "without perturbation" scenario. During training, the input tables are relevant to the question, so regardless of whether the gold query uses one or multiple tables, all input tables are used to predict the SQL. During inference, when irrelevant tables are added to the input, the model still tends to join those tables in the generated SQL. This demonstrates that the model learns spurious patterns, attempting to join all input tables due to only seeing relevant table schemas during training.

---

### Official Review · Reviewer_pQnW · 2024-11-03

**Soundness:** 3
**Presentation:** 3
**Contribution:** 2
**Rating:** 3
**Confidence:** 5

**Summary:**

The paper tackles the robustness issues of Text to SQL. Primarily, it focuses on how database schema influence the SQL generation by ML models/LLMs. The work called EVOSchema provides a in-depth study of how schema changes at each table level and column level affect the robustness of text to SQL. The study is performed by introducing an approach that augments training data with heuristic and LLM based schema perturbations and enabling the trained models to learn more effectively ignoring undesired patterns from the data. Experiments have been conducted using the most popular Text to SQL datasets SPIDER & BIRD to demonstrate the improvements with their approach. Overall, this is an important work that highlights the importance of robustness in SQL generation and also proposes a reasonable approach to tackle the database schema influence on the same.

**Strengths:**

1. Focused on a critical problem with LLMs and overall tackled it well.
2. This is a comprehensive framework for studying and improving robustness of Text to SQL against schema updates/changes.
3. Proposed metrics are robust and can serve as benchmark/baseline for further improvements in this specific area.
4. Training paradigm is diverse and well designed to assess the perturbations correctly.
5. Well articulated paper overall and easy to read.
6. Defined metrics are intuitive and appropriate for understanding the robustness.

**Weaknesses:**

1. First of all, the scope of the work is narrow i.e. it only focuses on one type of robustness challenge in Text to SQL.
2. The paper doesn't offer any novelty, originality or focuses on any strong theoretical foundations behind robustness issues in LLMs or any ML models.
3. There are several other works that tackle robustness and specific papers like ADVETA which tackle the robustness of SQL generation from schema changes. This work neither compares with them nor goes into detail about why this is one of the critical ways to tackle robustness in SQL.
4. Text to SQL systems don't necessarily use fine-tuned models. In such cases, this work is not relevant. Considering the generalizable nature of LLMs, it might be more effective in using such robustness strategies for post-training cases.

**Questions:**

There is just one big reason why this is not a sufficient contribution. The work is very narrowly scoped and offers very less novelty. These two can be improved well to make it a valuable contribution.

---

> ### Author Response · Authors · 2024-11-21
> **Author Response**
>
> We are pleased that the reviewer recognizes the significance of the problem we address, appreciates the comprehensiveness of the proposed framework, and finds the metrics we defined intuitive and well-suited for exploring this issue.
> ***
> [First of all, the scope of the work is narrow i.e. it only focuses on one type of robustness challenge in Text to SQL.]
>
> While we acknowledge that robustness can also encompass question-related aspects, it has already been comprehensively studied in [1]. Instead, our primary focus is on schema evolution, which is a critical and prevalent challenge in database design and maintenance. Robustness, in this context, is inherently linked to schema evolution. This problem is still underexplored though there is some work such as ADVETA [2] that tried to study this problem. They only explore two simple operations: “add columns” and “replace columns”, which very narrowly cover the schema evolution types in reality. We emphasize that delving deeper into such a significant problem and more comprehensively studying different schema evolution types is super important, as schema evolution is a frequent and underexplored issue. Our intentional focus on this problem, along with the proposed taxonomy,  in-depth analysis and training method, provides valuable insights into addressing the challenges of schema evolution.
> ***
> [The paper doesn't offer any novelty, originality or focuses on any strong theoretical foundations behind robustness issues in LLMs or any ML models.]
>
> * We comprehensively develop a taxonomy that covers both column-level and table-level schema evolution types to enable an in-depth study and present our findings, which provides a solid foundation for understanding how current models perform under various types of schema evolution and offering valuable insights into improving model performance to address such challenges.
> * There has been no exploration of robustness in modern models, such as the Llama family.
> * We build a unique way to synthesize training data to address the schema evolution challenge, which significantly enhance the robustness of the text-to-SQL models.
> ***
> [There are several other works that tackle robustness and specific papers like ADVETA which tackle the robustness of SQL generation from schema changes. This work neither compares with them nor goes into detail about why this is one of the critical ways to tackle robustness in SQL.]
>
> Thanks for the reviewer’s feedback. We have briefly described the differences between ADEVTA and our study in the related work (lines 518–519) and introduction (lines 80–81). ADEVTA primarily focuses on "add columns" and "replace columns," which represent a subset of the column-level schema evolution types covered in our work. In contrast, our study provides a more comprehensive and structured taxonomy, encompassing not only "add columns" and "rename columns" but also "remove columns." Additionally, for table-level schema evolution, we introduce types such as "add," "remove," "rename," "split," and "merge" tables. These evolution types are critical for real-world database updates but have not been systematically studied in previous work.
> ***
> [Text to SQL systems don't necessarily use fine-tuned models. In such cases, this work is not relevant. Considering the generalizable nature of LLMs, it might be more effective in using such robustness strategies for post-training cases.]
>
> Could you kindly clarify the distinction between post-training and fine-tuning? Additionally, could you provide a concrete explanation of what is meant by "using such robustness strategies for post-training cases"? Based on my understanding, the fine-tuning mentioned here appears to align with what you refer to as post-training.
> ***
> [1] Chang, et al. Dr.Spider: A Diagnostic Evaluation Benchmark towards Text-to-SQL Robustness. ICLR 2023
>
> [2] Pi, et al. Towards Robustness of Text-to-SQL Models Against Natural and Realistic
> Adversarial Table Perturbation. ACL 2022

---

> > ### Comment · Reviewer_pQnW · 2024-11-23
> > **Reply to Author's Response**
> >
> > Thanks for providing the clarification. I still don't generally agree with the novelty introduced here. I agree with the post-training vs fine-tuning having no technical distinction. In that case, this work acts as a synthetic data generation for improving the robustness of LLMs for text to sql.

---

### Official Review · Reviewer_vwxF · 2024-11-04

**Soundness:** 1
**Presentation:** 2
**Contribution:** 1
**Rating:** 3
**Confidence:** 4

**Summary:**

This paper addresses an important practical challenge in text-to-SQL systems: maintaining performance when database schemas evolve. A novel training paradigm was proposed to augment training data with diverse schema designs to improve model resilience. The experimental results reveal that table-level changes impact model performance more significantly than column-level changes, and their proposed training approach demonstrates substantial improvement over baseline models.

**Strengths:**

1. The introduction and related work sections are well organized. It clearly sketches the existing problem, the proposed argument, and related work for text-to-SQL.

2. The topic of this paper is interesting and practical since real-world databases are always evolving.

3. Intensive experiments have been conducted to verify the performance of the proposed model.

**Weaknesses:**

1. The method of dividing the training and testing sets is unclear in this paper.

2. How to get the new training and testing data with evolved databases is not clear. Do all the training and testing data share the same schema structure after the database evolves? If so, how can data leakage be avoided? If not, does it mean that there is only one change of database schema for each instance, and different instances share different structures of database schema?

3. Cost and Efficiency Analysis. This paper uses closed-source LLMs, like GPT-3.5 and GPT-4, to generate synthetic data. Is it expensive? How many budgets are needed to achieve a desirable performance?

4. Some experiment results are suspicious and meaningless. Table 1 presents comparative results between open-source LLMs and closed-source models (GPT-3.5 and GPT-4) under schema evolution scenarios. However, this comparison is fundamentally flawed since closed-source LLMs cannot be fine-tuned with evolved schemas. Without the ability to retrain or adapt these models, the performance comparison between scenarios with and without schema evolution lacks scientific validity. The authors should either remove these misleading comparisons or clearly explain their methodology for evaluating closed-source models in this context.

**Questions:**

See above.

---

> ### Author Response · Authors · 2024-11-21
> **Author Response**
>
> We are pleased that the reviewer finds our study of the problem both interesting and practical, appreciates the well-organized introduction and related work, and recognizes the comprehensive experiments that substantiate our claims.
> ***
> [The method of dividing the training and testing sets is unclear in this paper.]
>
> As lines 235-238 described in the paper, we follow the original training set and test set of the BIRD and Spider, and simulate the schema evolution based on the original training set and test set to make sure there is no overlap. Table 8 in Appendix A shows the statistics.
> ***
> [How to get the new training and testing data with evolved databases is not clear. Do all the training and testing data share the same schema structure after the database evolves? If so, how can data leakage be avoided? If not, does it mean that there is only one change of database schema for each instance, and different instances share different structures of database schema?]
>
> We use the original training sets and test sets of the BIRD and Spider. For the training set and test sets, we apply all the operations (add/remove/rename columns, add/remove/rename/split/merge tables) respectively. As there is no overlap or data leakage of the original training and test set, after we apply the schema evolution, there is still no overlap.
> ***
> [Cost and Efficiency Analysis. This paper uses closed-source LLMs, like GPT-3.5 and GPT-4, to generate synthetic data. Is it expensive? How many budgets are needed to achieve a desirable performance?]
>
> We utilized Azure OpenAI Service for data synthesis on BIRD, with costs of 30 dollars for GPT-3.5 and 480 dollars for GPT-4, totaling 510 dollars.
> ***
> [Some experiment results are suspicious and meaningless. Table 1 presents comparative results between open-source LLMs and closed-source models (GPT-3.5 and GPT-4) under schema evolution scenarios. However, this comparison is fundamentally flawed since closed-source LLMs cannot be fine-tuned with evolved schemas.]
>
> For closed-source models, they are not our major goal in our study. However, since closed-source LLMs are more powerful than open-source LLMs in general, and we are interested in how robust they are to the schema evolution. We mainly put the results there for reference and show some interesting findings.

---

### Official Review · Reviewer_rWbH · 2024-11-04

**Soundness:** 3
**Presentation:** 4
**Contribution:** 2
**Rating:** 6
**Confidence:** 4

**Summary:**

This paper has made some notable contributions as follow:
1. Contributed a new training method using augmented data in column-level and table-level to help Text-to-SQL model keep performance in the context of rapid changing schema in the future.
2. The new training approach also showed good results as the column-level and table-level change scenario given in the paper.

**Strengths:**

For originality: Paper introduces new appoarch to enhance performance of Text-to-SQL model. Paper also defines in detail the types of structural transformations that help standardize test cases and studies the stability of the text-to-SQL model such as: adding, removing, renaming for column-level and adding, removing, renaming, splitting, merging for table-level, laying the foundation for further research.

For quality: Creating augmented data by combining heuristics and GPT models ensures that the transformed data is reasonable, diverse, and of high quality. Furthermore, experiment results are clearly presented and demonstrate that the method actually improves the model results.

For clarity: Paper is presented in a logical, easy to understand with a clear structure. The goals and methods are described in detail.

For significance: Paper demonstrates that this approach is worthy of further testing and developing in the future.

**Weaknesses:**

The paper has a major flaw in the benchmark datasets and experiments:
1. The method only performs benchmarking on the dataset prepared by the author himself, there is no other dataset used to benchmark the data augmentation task in the paper. We need more external benchmark datasets.

2. The paper does not explore further fine-tuning methods to improve model performance. In addition, comparisons with closed-source models such as GPT models are still limited. For example, prompting methods for GPT models such as few-shot should be tried to compare with the models in the paper.

**Questions:**

1. Have you considered testing EvoSchema on larger, real-world databases beyond BIRD and Spider?

2.  Can you provide insights about heuristics methods for creating robust dataset?

3. Closed-source models are currently very high performing for code generation tasks like this. Why not consider tweaking them to make the answers of the closed-source models better so that comparing them with the results of a more meaningful method makes sense?

---

> ### Author Response · Authors · 2024-11-21
> **Author Response**
>
> We are grateful that the reviewer recognizes the originality, quality, clarity and significance of our paper.
> ***
> [The method only performs benchmarking on the dataset prepared by the author himself, there is no other dataset used to benchmark the data augmentation task in the paper.]
>
> BIRD and Spider are two of the most widely used open benchmarks in the text-to-SQL field, serving as the foundation for much of the existing research. And schema evolution is a significant and realistic problem frequently encountered in real-world scenarios yet remains underexplored. To address this, we formulate the schema evolution problem and introduce a new taxonomy covering both column-level and table-level schema evolution types. Since no existing dataset comprehensively captures all the schema evolution types required for our study, we constructed our benchmark by applying the proposed schema evolution types to these two popular open benchmarks.
> ***
>
> [The paper does not explore further fine-tuning methods to improve model performance. In addition, comparisons with closed-source models such as GPT models are still limited.]
>
> Our proposed training method is already fine-tuning the existing open-source models on our augmented training dataset (see all the w/ results in our experimental tables). And our augmented training data can encourage ​​the model to learn multiple schema designs and SQLs to the original question mappings, which can improve the model’s ability to identify the correct relationships among different tables and columns to the question, and can better distinguish the difference among different schema designs.
>
> For closed-source models, they are not our major goal in our study, so we don't try the few-shot prompting. We mainly put the results there for reference. Our study mainly explores how those open-source models are robust to the scheme evolution that frequently happens in the real world and the GPT results are just the reference.
> ***
>
> [Insights about heuristics methods for creating robust dataset]
>
> We take the renaming columns as an example, when renaming columns, the following heuristics should be applied to maintain consistency and avoid conflicts: 1) Preserve Meaning: The new column name should reflect the same meaning as the original column to avoid semantic confusion (e.g., CourseID → ClassIdentifier). 2) Avoid Conflicts: Ensure that the new column name does not conflict with existing column names within the same or other tables in the database. 3) Update References: Update all references to the renamed column in foreign keys in other tables. 4) Revise SQL: Update all SQL queries referencing the renamed column to work correctly after the renaming. These heuristics aim to ensure that renaming columns are performed systematically, maintaining the database's integrity and compatibility with SQL queries.

---

### Note · Authors · 2024-12-01

I have read and agree with the venue's withdrawal policy on behalf of myself and my co-authors.